# Isoleucine 44 Hydrophobic Patch Controls Toxicity of Unanchored, Linear Ubiquitin Chains through NF-κB Signaling

**DOI:** 10.3390/cells9061519

**Published:** 2020-06-22

**Authors:** Jessica R. Blount, Kozeta Libohova, Gustavo M. Silva, Sokol V. Todi

**Affiliations:** 1Department of Pharmacology, Wayne State University School of Medicine, 540 East Canfield St., Scott Hall Rm. 3108, Detroit, MI 48201, USA; jrblount@wayne.edu (J.R.B.); klibohov@med.wayne.edu (K.L.); 2Department of Biology, Duke University, Durham, NC 27708, USA; gustavo.silva@duke.edu; 3Department of Neurology, Wayne State University School of Medicine, 540 East Canfield St., Scott Hall Rm. 3108, Detroit, MI 48201, USA

**Keywords:** *Drosophila melanogaster*, innate immunity, NF-κB signaling, ubiquitination, unanchored poly-ubiquitin

## Abstract

Ubiquitination is a post-translational modification that regulates cellular processes by altering the interactions of proteins to which ubiquitin, a small protein adduct, is conjugated. Ubiquitination yields various products, including mono- and poly-ubiquitinated substrates, as well as unanchored poly-ubiquitin chains whose accumulation is considered toxic. We previously showed that transgenic, unanchored poly-ubiquitin is not problematic in *Drosophila melanogaster*. In the fruit fly, free chains exist in various lengths and topologies and are degraded by the proteasome; they are also conjugated onto other proteins as one unit, eliminating them from the free ubiquitin chain pool. Here, to further explore the notion of unanchored chain toxicity, we examined when free poly-ubiquitin might become problematic. We found that unanchored chains can be highly toxic if they resemble linear poly-ubiquitin that cannot be modified into other topologies. These species upregulate NF-κB signaling, and modulation of the levels of NF-κB components reduces toxicity. In additional studies, we show that toxicity from untethered, linear chains is regulated by isoleucine 44, which anchors a key interaction site for ubiquitin. We conclude that free ubiquitin chains can be toxic, but only in uncommon circumstances, such as when the ability of cells to modify and regulate them is markedly restricted.

## 1. Introduction

Post-translational modification adds a critical layer of complexity to the genome, allowing for regulation of many cellular functions. Methylation of histones in nucleosomes controls the expression of specific genes [1], phosphorylation of enzymes determines their activity [2], and glycosylation affects a protein’s conformation and ability to form complexes [3]. Ubiquitin (Ub) is a small protein adduct that is arguably known best for its role in protein quality control due to its participation in proteasome-dependent degradation [4]. Beyond that, Ub is an essential player in nearly all cellular processes, making it vital to eukaryotic cell health. A protein’s ubiquitination status can lead to various outcomes, from alteration of its protein–protein interactions to changes in its localization, function, enzymatic activity and half-life [5,6,7,8,9].

Ubiquitination is a sequential, ATP-dependent process that requires a Ub-activating enzyme (E1), a Ub-conjugating enzyme (E2), and a Ub ligase (E3) to covalently attach a Ub molecule to a substrate protein, typically at one of the substrate’s lysine residues [5]. With seven lysine residues of its own, Ub itself becomes ubiquitinated, resulting in the formation of poly-Ub chains with unique topologies dependent on the specific lysine residue linkers. Head-to-tail chains also exist, linked at the N-terminal methionine, as well as chains that contain a mixture of linkage types [10,11]. Ub chains that are not attached to a protein—known as unanchored, untethered, or free poly-Ub—can be synthesized by specialized E2s and E3s, and also arise from the removal of a preexisting chain from a ubiquitinated protein by deubiquitinating enzymes (DUBs) [12].

Ubiquitination is editable and reversible, ensuring that ubiquitinated substrates are targeted to the proper cellular complex while maintaining an available pool of Ub for reutilization. DUBs are a family of enzymes responsible for cleaving the linkages of Ub with its substrate or the linkages between Ub monomers in a poly-Ub chain. Depending on the DUB, it may influence the outcome for a ubiquitinated protein by editing the length or topological makeup of the Ub chain, or by removing the chain entirely [12,13,14]. In the case of unanchored poly-Ub, specific DUBs break down multiple, linked Ub molecules into mono-Ub; such is the case with Ubiquitin-Specific Protease 5, which has been evidenced to disassemble unanchored chains [15]. Processing of unanchored poly-Ub is important, as these chains participate in several processes, such as NF-kB signaling, and because disassembly of unanchored poly-Ub can replenish the pool of mono-Ub for reuse [14,16,17,18,19,20].

Unanchored poly-Ub has not been as well studied as other Ub species. There are several concepts regarding free poly-Ub based on studies in yeast and cultured mammalian cells: that it exists only at very low concentrations, that it is quickly disassembled by DUBs to be reused as mono-Ub, and that its accumulation is toxic [9,12,14,21,22]. It has been proposed that unanchored poly-Ub interferes with the normal function of the proteasome, either by directly interacting with and occluding it—i.e., free poly-Ub could begin its own degradation process, but overwhelms the proteasomal machinery—or by outcompeting ubiquitinated proteasomal substrates [23,24,25].

Our previous work introduced transgenic, DUB-resistant, untethered Ub chains into a physiological system, *Drosophila melanogaster*, to address questions surrounding the potential toxicity and handling of free poly-Ub in an intact, multicellular organism [26,27]. We synthesized DNA to express six Ub molecules in tandem, without any internal “GG” motifs (denoted as Ub^6^), which prevents them from being cleaved into mono-Ub by DUBs. Ub^6^ chains exhibited little-to-no toxicity, did not interfere with proteasomal function, and were readily subjected to proteasomal degradation [26]. These chains could also be transferred to a substrate protein en bloc in flies and in mammalian cells, without the need for prior disassembly into mono-Ub [26]. The DUB-resistant, free Ub^6^ chains become ubiquitinated themselves through various linkages, transforming them into mixed-linkage, free chains [26]. Additional work indicated that the continuous presence of these Ub chains in vivo does not cause marked changes at the transcriptome level in the fly [27]. Collectively, we interpreted these results to indicate that free poly-Ub is not necessarily detrimental in vivo and that Ub recycling may take place not only at the level of the monomer, but also through the reutilization of intact, untethered Ub chains.

Nonetheless, considering earlier studies that suggested toxicity from free chains [23,24,28], we sought to understand under what circumstances unanchored poly-Ub might become problematic to the cell. One key feature of the transgenic Ub chains expressed in flies is their conjugation with endogenous Ub [26]. This phenomenon made us wonder whether ubiquitination of these chains is important for the lack of toxicity we observe. Therefore, we generated new, isogenic fly lines that express Ub^6^ identical to the previously described species, with the exception that they lack all internal lysine residues, rendering them ubiquitination-resistant. The new chains essentially resemble free, linear poly-Ub. By manipulating the potential of Ub^6^ to be ubiquitinated, rather than the cell’s ability to ubiquitinate proteins, we avoid impacting ubiquitination processes more generally and can attribute changes in toxicity to the presence of Ub^6^.

We found stark differences in the toxicity of and response to ubiquitination-resistant Ub^6^ chains in vivo. Depending on the type of tissue where lysine-less Ub^6^ was expressed, it led to either developmental lethality or markedly shortened adult lifespan. Ubiquitination-resistant Ub^6^ has an extended turnover cycle when compared to Ub^6^ with all its lysine residues intact. Additionally, ubiquitination-resistant Ub^6^ leads to the upregulation of several components of NF-κB pathways in the fly, indicating a mechanism for toxicity from these free chains in *Drosophila*. Importantly, toxicity from ubiquitination-resistant Ub^6^ is reversed by mutating isoleucine (ile) residues at position 44 of each constituent Ub into alanine. Ile44 is a key part of a hydrophobic patch on Ub that controls many interactions [5,29,30,31,32]. We conclude that while unanchored poly-Ub is not inherently problematic in vivo, there can be circumstances in which these species may become detrimental, such as when cells’ abilities to manipulate and handle them are severely hampered. The studies that we describe here also introduce to the field new genetic tools for continued investigation of untethered poly-Ub.

## 2. Materials and Methods

### 2.1. Antibodies

Anti-HA (1:500 or 1:1000; rabbit monoclonal; Cell Signaling Technology, Danvers, MA, USA, #3724); anti-Ub (1:1000; rabbit polyclonal; Cell Signaling Technology, #3933); anti-tubulin (1:1000, mouse monoclonal, Sigma-Aldrich, St. Louis, MO, USA, #T5168); anti-Relish (1:500 from concentrate; mouse monoclonal; Developmental Studies Hybridoma Bank, Iowa City, IA, USA, #21F3); anti-lamin (1:1000; mouse monoclonal; Developmental Studies Hybridoma Bank, #ADL84.12); anti-phosphorylated-p65 (serine 536; 1:1000; rabbit monoclonal; Cell Signaling Technology, #3033); anti-p65 (1:1000; rabbit monoclonal; Cell Signaling Technology, #8242); goat anti-mouse, peroxidase-conjugated secondary (1:10,000; Jackson Immunoresearch, West Grove, PA, USA); goat anti-rabbit, peroxidase-conjugated secondary (1:10,000; Jackson Immunoresearch).

### 2.2. Fly Generation

GenScript (genscript.com; Piscataway, NJ, USA) synthesized Ub^6^ transgenes and cloned them into pWalium10.moe by using restriction sites EcoRI and XbaI, as also described before [26]. The amino acid sequences of the new lines generated for this study are shown in Figure 1 and Figure 5. The purified plasmid was injected into yw; attP40 by Duke University Model Systems, using the PhiC31-targeted integration to achieve site-specific, unidirectional integration into the second chromosome [33,34]. The transformed chromosome was migrated onto the w^1118^ parental line and balanced lines were confirmed through genomic DNA-based PCR assays for transgene insertion and orientation, as described before [35,36]. All fly lines were sequence verified.

### 2.3. Drosophila Stocks and Procedures

All flies were maintained in diurnal incubators at 25 °C and ~60% humidity, in conventional cornmeal media. Where noted, RU486 was added to the conventional media, as previously described [37]. Tubulin-Gal4-GS was a generous gift from Dr. R.J. Wessells, Wayne State University; sqh-Gal4 was a gift from Dr. Daniel Kiehart, Duke University; repo-Gal4 was gifted by Dr. Daniel Eberl, University of Iowa. Mef2-Gal4 (#27390) and elav-Gal4 (#458) were from Bloomington *Drosophila* Stock Center (Bloomington, IN, USA). The following RNAi lines were also from the Bloomington *Drosophila* Stock Center: *Relish* (#3361), *Dif* (#30513), *cactus* (#31713), *dorsal* (#27650), *IKKBeta* (#35186), *kenny* (#35572, 57759). *dorsal* overexpression line was from FlyORF (Zurich, Switzerland, #000638). For longevity assays, male and female flies were collected as they eclosed from their pupal cases and aged on conventional cornmeal fly media at 25 °C, with ~20 flies per vial. Flies were transferred to fresh vials every 2–3 days until all were dead. Throughout this study, flies were heterozygous for driver and transgenes.

### 2.4. Western Blotting

Five whole adults flies or pharate adults, or ten dissected heads per group were lysed as described previously [26]. Blots were developed and quantified using a CCD-equipped VersaDoc system and Quantity One software (Bio-Rad, Hercules, CA, USA; version 4.6.8), Syngene PXi4 and GeneSys software (Syngene, Frederick, MD, USA; version 1.7.2), or ChemiDoc and ImageLab (Bio-Rad; version 6.0.1).

### 2.5. Immunoprecipitation and In Vitro Deubiquitination

For precipitation of HA-Ub^6^ from flies, 40 flies per group were homogenized in RIPA lysis buffer (50 mM Tris, 150 mM NaCl, 0.1% SDS, 0.5% deoxycholic acid, 1% NP40, pH 7.4) supplemented with protease inhibitor cocktail (PI; Sigma-Aldrich), sonicated, centrifuged at 15,000× *g* for 20 min at 4 °C, and then incubated with anti-HA antibody-bound beads for four hours, tumbling at 4 °C. Beads were rinsed 10 times (including 2 × 5 min tumbling at 4 °C), and then split into two microfuge tubes and incubated at 37 °C with either NEM (0.5 mM) or the catalytic domain of USP2 (0.1 µM; Boston Biochem, Cambridge, MA, USA). After incubation for the selected times, reactions were stopped by the addition of 6% SDS sample buffer and by boiling for 1 min.

### 2.6. Soluble/Insoluble Fractionation

Five flies per group were mechanically homogenized in 200 μL of NETN lysis buffer (50 mM Tris pH 7.5, 150 mM NaCl, and 0.5% IPEGAL ca-630) supplemented with PI, sonicated briefly, and then centrifuged at 20,000× *g* for 30 min at 4 °C. The resulting supernatant was removed, transferred into a new microfuge tube, and subsequently quantified by using the BCA assay (ThermoFisher, Waltham, MA, USA). The remaining pellet was fully resuspended in 200 μL of PBS + 1% SDS by vortexing and boiling. Then, 10 μg of the supernatant fraction and 7 μL of the pellet fraction were supplemented with loading buffer, boiled briefly and electrophoresed on SDS-PAGE gels for analyses by Western blotting.

### 2.7. Quantitative Real-Time PCR

From 10 dissected pharate heads per group, total RNA was extracted using TRIzol reagent (Invitrogen, Carlsbad, CA, USA) and treated with TURBO DNase (Ambion, Austin, TX, USA) to eliminate contaminating DNA. A high-Capacity cDNA Reverse Transcription Kit (ABI, Waltham, MA, USA) was used to obtain a cDNA library, and preamplification of the genes of interest was performed using SsoAdvanced Preamplification Supermix (Bio-Rad). A Fast SYBR Green Master Mix (ThermoFisher) was used in conjunction with a StepOne real-time PCR system. mRNA levels were quantified using 2^−ΔΔCt^ (cycle threshold) methods and normalizing all transcripts to the reference gene, *rp49*. The following primer sequences were used: 

*rp49* (F: AGATCGTGAAGAAGCGCACCAAG; R: CACCAGGAACTTCTTGAATCCGG), *Relish* (F: ATGAACTTGAACCAGGTGCGG; R: TGCCGACTTGCGGTTATTGA), *Dif* (F: CAGAGTTCCAACCCACGGAC; R: AGGAGTTCTGGATTCGGGTAGT), *dorsal* (F: AGAGCCCGCCAAGGTTTTT; R: TGCCATCCTGGTGGTCATTC), *cactus* (F: ATCCAACGACAAAGCGGTCA; R: GATTTTCCCTCCCTGGCGTTA), and *kenny* (F: TACCTCGCGCTAAAGAGCAC; R: CAGCTCTTGGTTTTCCACGC).

### 2.8. Subcellular Fractionation

A ReadyPrep Cytoplasmic/Nuclear Protein Extraction Kit (Bio-Rad) was used to separate nuclear and cytoplasmic proteins from 10 pharate adult dissected heads, or individual wells in a 24-well culture plate for experiments with HEK-293T cells. The manufacturer’s protocol was used, with two variations: (1) nuclear fractions were rinsed once with cytoplasmic protein exclusion buffer (CPEB) for 10 min, tumbling at 4 °C before subsequent incubations in CPEB on ice; (2) cytoplasmic fractions were centrifuged 3×, with the supernatant transferred to a fresh tube between spins.

### 2.9. Mammalian Cell Maintenance and Transfection

HEK-293T cells were maintained under conventional conditions in DMEM with 10% FBS and 5% Penicillin-Streptomycin. Cells were transfected using Lipofectamine LTX (Invitrogen), using the manufacturer’s protocols. Twenty-four hours after transfection, cells were collected and rinsed in cold PBS, and then used for whole-cell lysis (using the same buffer and protocol as for blots with flies), or for subcellular fractionation. DNA constructs used for transfections were the same as those synthesized for fly generation, but subcloned into the mammalian vector pcDNA3.1.

## 3. Results

### 3.1. New Drosophila Lines Expressing Ubiquitination-Resistant, Unanchored Chains

As introduced above, we previously examined whether expression of unanchored poly-Ub is toxic in *Drosophila.* Using the Gal4-UAS system [38,39], we expressed Ub^6^ chains that cannot be dismantled by DUBs (e.g., Ub^6^-Stop in Figure 1A). These poly-Ub resemble linear, untethered chains that exist in nature [5,14,21]. The chains were constructed to either remain unanchored, due to the lack of the terminal “GG” motif that would enable their isopeptide-bond fusion to another protein (Ub^6^-Stop; Figure 1A), or could be tethered wholesale onto another protein as a result of a single “GG” motif at the very end [26]. We found that these Ub^6^ species are innocuous [26,27]. Nonetheless, as some studies concluded that untethered poly-Ub is toxic [23,24,28], we sought to explore when free chains might become detrimental to the cell. We focused on a potential role from the ubiquitination of untethered poly-Ub by endogenous Ub, since this is a major feature of transgenically-expressed chains [26].

We designed head-to-tail Ub^6^ that, similar to the prior constructs [26], does not contain “GG” motifs internally or at the very end, and is also devoid of lysine residues (lys to arg mutation), making it resistant to ubiquitination (Ub^6^-Stop-K0; Figure 1A). Removing the potential for ubiquitination of Ub^6^ allows us to examine the behavior of an untethered Ub chain that cannot be decorated by ubiquitination within the cell, in contrast to the mixed-linkage chains that arose and were highly abundant in our previous studies [26].

We generated new transgenic flies that incorporate Ub^6^-Stop-K0 at the same chromosomal site that contains Ub^6^-Stop, attP40 on the second chromosome of *Drosophila*. The PhiC31 system of transgene integration ensures single copy, same orientation insertion of the intended construct into the same chromosomal site [26,33,34,40,41,42]. Following established protocols from our published work [35,36], we utilized PCR-based assays and sequencing to ensure integration of the transgene into the right locus and in the correct orientation, and to ensure that the sequence inserted was the one intended (data not shown). With these isogenic lines on hand, we began our investigations. To characterize the biochemical properties of Ub^6^-Stop-K0 in comparison to Ub^6^-Stop, we first performed Western blotting using lysates collected from flies expressing either version of Ub^6^. We observed Ub^6^-Stop-K0 at the expected molecular weight of ~50 kDa, without the distinct and clearly defined ubiquitination laddering that we see above Ub^6^-Stop (Figure 1B and other supportive information in data collected for Section 3.3 and Section 3.5). At this point, it is interesting to note that the precise type of ubiquitination laddering above the main band of Ub^6^-Stop differs among different tissues. In muscle cells, there appears to be a higher preponderance of >Ub^1^ addition (Ub^3–5^, middle portion of the blots in Figure 1B; see also data pertaining to Section 3.5.) compared to neuronal and glial cells, whose signal is stronger closer to the main, unmodified band. The reasons for these differences and their potential physiological consequences are unknown. Still, the key point from these comparisons is that Ub^6^-Stop-K0 migrates similarly in different fly tissues.

We showed previously that the laddering above the main band of Ub^6^-Stop comprises ubiquitinated species [26]. One of the methods we utilized was to incubate immunoprecipitated (IP-ed) Ub^6^-Stop with the catalytic domain of USP2 (USP2_CD_), a DUB that would remove Ub added onto Ub^6^ in any type of linkage, including met1/linear fusions [43], while leaving Ub^6^ itself intact due to absence of “GG” motifs. As shown in Figure 1C, while USP2_CD_ readily removes conjugated Ub from Ub^6^-Stop, as evidenced by the collapse of the higher molecular weight (HMW) bands into the main Ub^6^ band, it has no detectable effect on Ub^6^-Stop-K0. The faint HMW smear observed in the long-exposure image in Figure 1C is likely aggregated Ub^6^-Stop-K0, as suggested by centrifugation protocols (Figure 1D). Through centrifugation of fly lysates, we observe the presence of some Ub^6^-Stop, and especially of Ub^6^-Stop-K0, in the pellet fraction, suggesting that both chains may aggregate (Figure 1D). Together, these data indicate that Ub^6^-Stop-K0 is expressed in the fly and that it is resistant to ubiquitination.

### 3.2. Lysine-Less, Unanchored Poly-Ub Is Highly Toxic in Drosophila

We found before that expression of Ub^6^-Stop in all tissues or only in muscle cells did not impact longevity, while restricted expression in glial cells or neurons reduced adult fly longevity slightly, by a few days (normal flies can live for approximately 90 days; [26]). Here, we used various Gal4-UAS drivers to test the impact of Ub^6^-Stop and Ub^6^-Stop-K0 in fly longevity. sqh-Gal4 is a commonly used driver that leads to expression of the transgene of interest in all tissues during development and as adults [26,40,41,44,45]. Flies heterozygous for sqh-Gal4 and Ub^6^-Stop-K0 die as embryos or young larvae (Figure 2A). elav-Gal4 (pan-neuronal expression) and repo-Gal4 (pan-glial expression) both lead to developmental death and drastically shortened adult lifespans when expressing Ub^6^-Stop-K0 (Figure 2A–C). In the case of glial expression, no Ub^6^-Stop-K0-expressing adults emerged from most crosses that we set up; developing flies died as pharate adults. We were able to collect some live adults in later crosses for experiments described in Section 3.4 and their lives were significantly shorter than control adults. It is the longevity of these adults that is shown in Figure 2C. Expression of Ub^6^-Stop-K0 in muscle cells (Mef2-Gal4) results in a less dramatically, but still statistically significantly abbreviated lifespan (Figure 2A,D). In essence, with every Gal4 driver used, Ub^6^-Stop-K0 leads to markedly reduced lifespan compared to both Ub^6^-Stop flies and control flies that are genetically similar but do not express either Ub^6^ (Figure 2). Clearly, different drivers lead to variable toxicity from ubiquitination-resistant, linear poly-Ub. Among the tissues examined, muscle cells are least impacted. This is unlikely to be directly due to differences in relative amounts expressed by the different drivers, since we showed before that muscle cells express Ub^6^ transgenes highly robustly [26]. More likely, different tissues handle and tolerate these species in different ways, leading to the observed variation in toxicity. These findings highlight the need to investigate Ub-dependent pathways in a tissue-specific manner to understand the full scope of their roles and importance in vivo. Collectively, these data point to marked toxicity from ubiquitination-resistant, untethered Ub^6^ species in *Drosophila*, unlike the ubiquitination-prone counterpart.

### 3.3. Lysine-Less, Unanchored Poly-Ub Is More Stable in the Fly

Since endogenous ubiquitination may influence the toxicity of Ub^6^, we questioned whether fly cells clear Ub^6^-Stop-K0 as readily as Ub^6^-Stop, which can be ubiquitinated. Lys48- and lys11-linked ubiquitination are proteasomal degradation signals [4,5,46,47] that are present on Ub^6^-Stop ([26] and unpublished observations from J. R. Blount and S. V. Todi). It is not unreasonable to hypothesize that their absence could hinder the degradation of Ub^6^-Stop-K0. To determine whether Ub^6^-Stop-K0 is turned over more slowly than Ub^6^-Stop, we employed the RU486-dependent, ubiquitous driver, tub-Gal4-GS to perform an experiment akin to a pulse-chase [26]. Flies heterozygous for tub-Gal4-GS and either form of Ub^6^ were reared on regular fly food until they emerged from their pupal cases, at which point they were switched to RU486 food for 2 days to induce Ub^6^ transgene expression (the “pulse”). Adults were then switched to regular media to halt additional Ub^6^ production and were flash-frozen periodically to assess Ub^6^ disappearance by Western blotting (the “chase”). We selected an inducible, ubiquitous Gal4 driver and focused on adult flies to gain a general perspective on the turnover of these Ub species, aware of the fact that degradation rates may differ in a tissue- and developmental stage-dependent manner. Nonetheless, this approach provides us with critical insight into the handling of ubiquitination-prone and ubiquitination-resistant, linear poly-Ub at the level of the whole organism.

Figure 3 summarizes our findings. We observe that the lack of ubiquitination stabilizes Ub^6^-Stop-K0 compared to Ub^6^-Stop. Fifty percent of Ub^6^-Stop-K0 is degraded by approximately 72 h in comparison to less than 24 h for Ub^6^-Stop. By 7 days, both proteins reach comparable levels. These data suggest that internal lysine residues and their ubiquitination are critical, but not entirely essential for chain turnover in flies.

### 3.4. Toxicity of Lysine-Less, Unanchored Poly-Ub Depends on NF-κB Signaling

Recent studies focusing on innate immunity pathways have revealed roles for free poly-Ub in NF-κB signaling cascades [16,48,49,50,51,52]. “NF-κB” refers to a family of inducible transcription factors which, in response to the activation of immune receptors by invading microbes, induce the expression of antimicrobial peptides. In *Drosophila*, innate immunity is primarily regulated by two pathways that activate NF-κB: Imd and Toll (Figure 4A). During the *Drosophila* immune response mediated by NF-κB pathways, a series of downstream regulators are involved (e.g., cactus, dorsal, kenny, and Relish; Figure 4A). Lysine-less Ub^6^ resembles unanchored, linear chains that have been proposed to regulate components of NF-κB in the mammalian system [53,54,55]; therefore, we posited that Ub^6^-Stop-K0 might mimic endogenous poly-Ub involved in the Imd pathway and cause abnormal signaling, which could explain toxicity that we observe with the expression of Ub^6^-Stop-K0 chains (Figure 2).

To address the possibility of aberrant NF-κB signaling in the toxicity of lysine-less Ub^6^, we used the repo-Gal4 driver to express in glia Ub^6^-Stop-K0 alongside RNAi transgenes targeting components of the *Drosophila* Imd and Toll pathways, and monitored effects on longevity. We chose repo-Gal4 because Ub^6^-Stop-K0 expression in glial cells causes toxicity in developing and adult flies (Figure 2A,C); thus, any relief of this toxicity is observable along developmental and adult stages.

Independent knockdown of *Relish* and *Dif* significantly alleviates toxicity from Ub^6^-Stop-K0, although neither restores adult fly lifespan to that of the controls that do not express Ub^6^-Stop-K0 (Figure 4B). A separate longevity study revealed that knockdown of *dorsal*, *cactus* and *IKKβ* also significantly reduces toxicity (Figure 4C). In this second study, Ub^6^-Stop-K0 leads to widespread pharate adult lethality with no surviving adults (Figure 2A summarizes fly stages of development). We have observed that transient factors, such as variation in ingredient lot numbers, can influence the development and survival of these flies; therefore, all comparisons in development and longevity are made only among flies reared simultaneously. Still, even though we notice some variation in the extent of toxicity, Ub^6^-Stop-K0 is consistently toxic and knockdown of NF-kB components consistently leads to increased survival (Figure 4B,C). The one exception is kenny. Knockdown of *kenny* does not improve toxicity from Ub^6^-Stop-K0. No flies with pan-glial expression of both *kenny* RNAi and Ub^6^-Stop-K0 survive to adulthood, despite the eclosure of a small number of flies expressing Ub^6^-Stop-K0 alone (Figure 4B). This observation indicates that the toxicity-rescuing effects we see with NF-κB knockdown are specific to key genes. In summary, targeting of several NF-κB players reduces the toxicity of lysine-less, free Ub^6^, implicating Ub^6^-Stop-K0 in immune signaling via both Imd and Toll pathways.

To further explore this implication, we performed qRT-PCR using dissected heads from pharate adults expressing Ub^6^-Stop-K0 and repo-Gal4 to drive expression in glial cells only, compared to those with the same genetic background but without the Ub^6^ transgene. We used pharate adult heads because most Ub^6^-Stop-K0 adult flies do not eclose and because *repo* is robustly expressed in the head (flyatlas.com). In the presence of Ub^6^-Stop-K0, we observe statistically significantly increased levels of *Relish*, *Cactus*, *Dif*, and *dorsal* mRNA. In agreement with our previous results, *kenny* mRNA levels trend upward, however without reaching significance (Figure 4D). *dorsal* mRNA, in particular, is markedly upregulated. Importantly, overexpression of *dorsal* in glial cells recapitulates lethality in flies that do not express any Ub^6^, indicating that *dorsal* upregulation alone is sufficient for toxicity (Figure 4E). These results illustrate a disturbance in normal NF-κB signaling in the presence of Ub^6^-Stop-K0 and present a mechanism for unanchored chain toxicity that relies on aberrant NF-κB upregulation.

### 3.5. Aberrant NF-κB Signaling Is Mediated by the Ile44-Centered Hydrophobic Patch of Lysine-Less, Linear Poly-Ub

Proper folding of Ub molecules results in a hydrophobic patch centered around amino acids isoleucine (ile) 44, leucine 8, histidine 68, and valine 70 (Figure 5A; [5]). This surface is the recognition site for many Ub-binding domains (UBDs) and is thus essential to the interaction of Ub with other proteins and for many of its actions (Figure 5B; [30,31,32,56]). NEMO, the NF-κB regulator, binds linear and lys63-linked chains [54,57] and 3D analysis of this interaction shows a critical position for Ub’s ile44 (Figure 5C). Likewise, TAB2 and TAB3, which activate TAK1 in mammalian NF-κB signaling, recognize the ile44 patch through their C-terminal Np14 zinc finger domains [58]. 

Reasoning that ile44 within Ub^6^-Stop-K0 is likely important for its incorporation into, or recognition by, NF-κB signaling, we generated an additional Ub^6^ chain identical to Ub^6^-Stop-K0, with the exception that every ile44 residue is mutated to alanine (Ub^6^-Stop-K0-ile44a; Figure 5D), a mutation that is commonly used to disrupt the hydrophobic patch [30,31,32]. These new chains allow us to examine the influence of the ile44 hydrophobic patch on the toxicity of Ub^6^ chains. Expression of Ub^6^-Stop-K0-ile44a in *Drosophila* muscle cells and in glial cells leads to protein levels that are comparable to those of Ub^6^-Stop-K0. Similar to Ub^6^-Stop-K0, Ub^6^-Stop-K0-ile44a does not show the distinct HMW bands consistent with ubiquitination in Western blots. This is contrary to Ub^6^-Stop, which presents with clear and distinct laddering of ubiquitination above its main band (Figure 5E).

To assess toxicity from Ub^6^-Stop-K0-ile44a, we used the tissue-specific drivers described earlier to express the new transgene and then performed longevity assays. In each case, Ub^6^-Stop-K0-ile44a flies survive longer than their Ub^6^-Stop-K0 counterparts (Figure 6A,B). In fact, Ub^6^-Stop-K0-ile44a expression in muscle cells and in neurons does not have a statistically significant effect on adult fly lifespan compared to controls that express no Ub^6^. Driving expression in all fly cells or only in glia results in a modest reduction in longevity (Figure 6B). It is particularly remarkable that disrupting the hydrophobic interaction interface of Ub eliminates developmental lethality from ubiquitous expression of Ub^6^-Stop-K0 and only mildly reduces adult fly longevity compared to controls without Ub^6^ (Figure 6A,B).

Next, we examined whether reduced toxicity from Ub^6^-Stop-K0-ile44a coincides with normalized levels of genes involved in NF-κB signaling, compared to Ub^6^-Stop-K0. We recapitulated studies conducted for Figure 4D by using new crosses and adding flies that express Ub^6^-Stop-K0-ile44a in all glia. As summarized in qRT-PCR results in Figure 6C, the mRNA levels of *dorsal* and *Relish* are not statistically different in the presence of Ub^6^-Stop-K0-ile44a compared to flies not expressing any Ub^6^ transgenes, but continue to be upregulated in the presence of Ub^6^-Stop-K0.

We validated these findings by examining the protein levels of Relish (antibodies tested for other NF-κB components in flies were non-specific or inconsistent). We observed that in the presence of Ub^6^-Stop-K0, Relish protein levels are noticeably higher, compared to flies not expressing any Ub^6^ transgene, or expressing the ubiquitinatable Ub^6^-Stop. In the presence of Ub^6^-Stop-K0-ile44a, Relish levels return towards normality (Figure 6D). Because NF-κB transcription factors translocate to the nucleus when activated, we performed subcellular fractionation to visualize nuclear Relish protein levels in heads dissected from pharate adults expressing Ub^6^ in glial cells. We observed that more Relish is present in the nuclei of Ub^6^-Stop-K0 pharate adults, compared to controls that are genetically similar, but lack Ub^6^ (Figure 6E). Nuclear Relish levels in the presence of either Ub^6^-Stop or Ub^6^-Stop-K0-ile44a appear lower than those in the presence of Ub^6^-Stop-K0, indicating the importance of the chains’ susceptibility to ubiquitination and ile44. Collectively, these results lead us to conclude that ubiquitination-resistant, free, linear Ub chains can be highly toxic in flies at least in part due to NF-κB signaling that is dependent on the ile44-centered hydrophobic patch on Ub.

Due to a lack of quality antibodies for NF-κB components in flies, we performed subcellular fractionation experiments using cultured, HEK-293T human cells to validate our findings in Figure 6E. In humans, p65 is an NF-κB transcription factor whose phosphorylation at serine 536 controls its nuclear translocation and activity [59]. To examine p65 translocation, we transiently transfectedHEK-293T cells with the Ub^6^ constructs and, 24 h later, performed subcellular fractionation and Western blotting. Expression of the various Ub^6^ constructs in HEK-293T cells leads to ubiquitinated species for Ub^6^-Stop, but not so for lysine-less variants (Figure 7A). As shown in Figure 7B, Ub^6^-Stop-K0 expression leads to markedly more endogenous phosphorylated-p65 in the nucleus compared to cells transfected with empty vector; the ile44a mutation reverses this effect. There was no statistically significant difference in the total p65 levels in the cytoplasm or nucleus (Figure 7B). Based on the increased nuclear translocation of activated p65, it is likely that Ub^6^-Stop-K0 induces the transcription of downstream NF-κB response genes; additional studies are needed to determine the precise outcomes. These results reinforce a role for ubiquitination-resistant, unanchored, linear poly-Ub in NF-κB signaling, notably in a human cell line.

## 4. Discussion

We have developed a model to study the control and function of untethered poly-Ub in an intact, multicellular organism, *Drosophila melanogaster*. By manipulating the properties of linear, unanchored poly-Ub, we propose that free chains are not intrinsically toxic, but that the ability of cells to control them, e.g., through endogenous ubiquitination, is crucial to their remaining innocuous. Ub^6^-Stop-K0, which is unreceptive to ubiquitination, causes aberrant NF-κB signaling and fly death that is largely dependent on amino acid ile44 on Ub. Our results, therefore, also demonstrate a critical role for ile44 on Ub in NF-κB-dependent pathways.

The Ub^6^ chains that we designed are head-to-tail by necessity; current technology limits the ability to stably, genetically encode other linkage types in vivo. Endogenous, unanchored poly-Ub chains are diverse in their linkage types and lengths, and their composition controls their functions and interaction partners [5,10,12,16,51]. We previously characterized Ub^6^ species that are endogenously ubiquitinated, transforming them into branched chains with various lysine linkages [26]. Unlike them, Ub^6^-Stop-K0 and Ub^6^-Stop-K0-ile44a that we described here are homogenous, linear poly-Ub. We also established in prior work [26] and through the results of this study that each of the Ub^6^ species has distinct properties, from turnover rates to physiological outcomes. For these reasons, the transgenically-encoded Ub^6^ fly lines that we developed serve as valuable tools to study the function and regulation of unanchored poly-Ub in vivo.

It remains to be resolved which proteins recognize and interact with Ub^6^-Stop-K0. For example, upstream elements that lead to upregulated NF-κB component levels in the presence of lysine-less, linear poly-Ub need to be identified. Still, we infer that these elements likely depend on interacting with the ile44 hydrophobic patch, since mutating this amino acid residue markedly reverses Ub^6^-Stop-K0 toxicity in intact flies. Our results highlight NF-κB components as key players in toxicity from ubiquitination-resistant, linear chains. According to mammalian studies, canonical activation of NF-κB requires the linear Ub chain assembly complex (LUBAC). LUBAC attaches head-to-tail, met1-linked chains to NEMO, an NF-κB regulator, and participates in the synthesis of branched, unanchored poly-Ub chains composed of both met1- and lys63-linkages [16]. Free Ub chains are required for the activation of the MAP kinase kinase kinase, TAK1 and the subsequent phosphorylation and degradation of the inhibitory kinase, IKKβ [50]. Several other proteins within the pathway, including NEMO and the LUBAC proteins, contain UBDs that may interact with untethered chains and may be regulated by them [49]. Thus, it is likely that the presence of lysine-less, linear, unanchored poly-Ub places NF-κB in overdrive by interacting with NEMO and LUBAC. In the absence of a functional binding interface on Ub (the ile44a mutation), NF-κB is not improperly triggered. Our results lead to the conclusion that toxicity induced by linear chains is due to overstimulation of the NF-κB pathway. In the case of the ile44a mutant, these linear chains are not recognizable by the cellular components so their expression *per se* does not induce toxicity.

As introduced earlier, flies have two NF-κB pathways, Imd and Toll. The Imd pathway is initiated when Gram-negative bacteria are detected by peptidoglycan recognition protein-LC at the plasma membrane [60,61,62]. This event triggers the activation of Tab2/Tak1 that phosphorylates and activates the IKK complex (kenny and IKKβ) [63,64,65] which then, via Relish (the NF-κB transcription factor), activates antimicrobial genes including Diptericin and Cecropin ([66,67,68] and Figure 4A). The involvement of unanchored poly-Ub in the Imd pathway is expected at multiple steps, based on mammalian NF-κB signaling orthologues. Ubcd4, an E2 conjugase whose mammalian orthologue, E2-25K/Ube2K, generates poly-Ub chains without a substrate protein [69], is required for Imd activation [70]. The concomitant knockdown of the E3 ligase Bendless and the E2 conjugase Uev1a abrogates Imd signaling in cultured insect cells [71], and Bendless/Uev1a together synthesize unanchored lys63-linked Ub chains in vitro [72]. To the best of our knowledge, there are no published reports showing a direct involvement of unanchored poly-Ub in the Imd pathway in *Drosophila*, but the importance of Ubcd4 and Bendless/Uev1a implicates these chains. Our results that linear poly-Ub activates *Relish* strengthen this connection and further indicate that free, linear Ub chains are involved in the Imd pathway in flies.

Unlike for the Imd pathway, where evidence from mammalian orthologues and functional assays of fly proteins indicates a role for free Ub chains, it is unclear whether unanchored poly-Ub is involved in the Toll pathway. Toll signaling begins when Gram-positive bacteria or fungi are detected by extracellular peptidoglycan recognition proteins and glucan-binding proteins, triggering a cascade that results in the processing of the Toll ligand Spaetzle to facilitate its binding to the Toll receptor [73,74,75]. After activation, the Toll receptor binds the adaptor MyD88 and recruits Tube and Pelle to phosphorylate cactus [76], ultimately activating dorsal and Dif [77] and driving the transcription of antimicrobial peptides like Drosomycin and Defensin ([78] and Figure 4A). Insofar as we know, our data that free, linear poly-Ub increases levels of *cactus* and *dorsal* are the first to link linear chains to the Toll pathway, thus merging Gram-negative and Gram-positive/fungi, NF-κB-centered processes under the regulation of linear Ub chains. Which proteins serve as “receptors” that recognize lysine-less, linear poly-Ub and lead to activation of the Toll pathway? Studies with unbiased approaches and targeted genetics will be necessary to uncover these details and also to identify downstream genes that are impacted by free, lysine-less Ub chains for both the Imd and Toll pathways. Based on increased nuclear levels of Relish and phosphorylated-p65 proteins in flies and mammalian cells expressing Ub^6^-Stop-K0, there is good reason to predict a genetic response to these transcription factors.

The extent of NF-κB component perturbation by linear Ub chains differed among experiments in this work. For example, in Figure 6C, glial expression of Ub^6^-Stop-K0 did not induce *dorsal* mRNA expression to the magnitude shown in Figure 4D. As mentioned above, we have observed physiological effects from different fly media conditions, even using the same food recipe. Flies used in Figure 4 were reared on a lot of food that was different from those in Figure 6, which might have been more prone to mold or bacterial growth and could have itself induced background NF-κB signaling. We stress the importance of making comparisons only among flies and their respective controls that were maintained on the same batch of food and monitored at the same time. In our case, all flies from Figure 4D were reared together and collected independently of those in Figure 6C, which were reared and collected as a separate cohort. However, regardless of the observed experimental variation, the important point remains that the induction of both *dorsal* and *Relish* mRNA was consistent among flies reared simultaneously: Ub^6^-Stop-K0 significantly increases their levels, whereas Ub^6^-Stop-K0-ile44a does not have the same impact.

Our work provides new insight into Ub biology more generally. The current model of Ub utilization and recycling centers on the use of mono-Ub in a stepwise manner to generate poly-Ub and the disassembly of poly-Ub into mono-Ub before reuse (Figure 8). We previously described [26] and further evidence here the possibility of alternative routes for Ub use and recycling, based on the notion that free Ub chains may not need to be disassembled into mono-Ub to be eliminated. They can be conjugated en bloc onto other proteins, or can be degraded without the need of disassembly into their building blocks [26]. These routes can effectively recycle untethered poly-Ub, which seems largely innocuous to an intact organism, except under highly specific conditions, such as the ones in the current study (Figure 8). We propose that cells might employ various types of Ub utilization, recycling and disposal routes, depending on homeostatic needs. For example, conjugation of free poly-ub onto specific proteins without prior disassembly might be beneficial under times of energy stress or proteotoxic pressure, where quick and efficient removal of certain proteins might benefit from Ub chain “hopping”. These and other possibilities deserve attention to understand Ub biology.

*Drosophila* is an excellent model organism due its flexible genetics, easy and inexpensive maintenance, low number of chromosomes, and short life cycle. Importantly, most genes and pathways are conserved among *Drosophila* and vertebrates, allowing for reliable translation of findings from flies to higher-order organisms [79]. It will be important to expand our findings in a mammalian system to examine the role of endogenous, unanchored poly-Ub in various disease models and stress responses. Free Ub chains are upregulated under certain stress conditions, including inflammation [18]—what types of unanchored chains might we see in mammals with inflammatory diseases? Are there tissue-specific differences? Are there disease-specific variations? Diversity in unanchored Ub chain signaling—from chain length to composition to available binding interfaces—will be important to consider when studying the effects of these species going forward.

In summary, our results reinforce the notion that free Ub chains are not inherently detrimental. Our investigations contribute to the overall understanding of Ub biology, recognize the ile44-centered hydrophobic patch of Ub as an important site for toxicity, and identify ways to reduce and prevent deleterious effects from untethered Ub chains in vivo (Figure 8). Continued work is necessary to probe into these possibilities in different organisms and under various physiological conditions.

## Figures and Tables

**Figure 1 cells-09-01519-f001:**
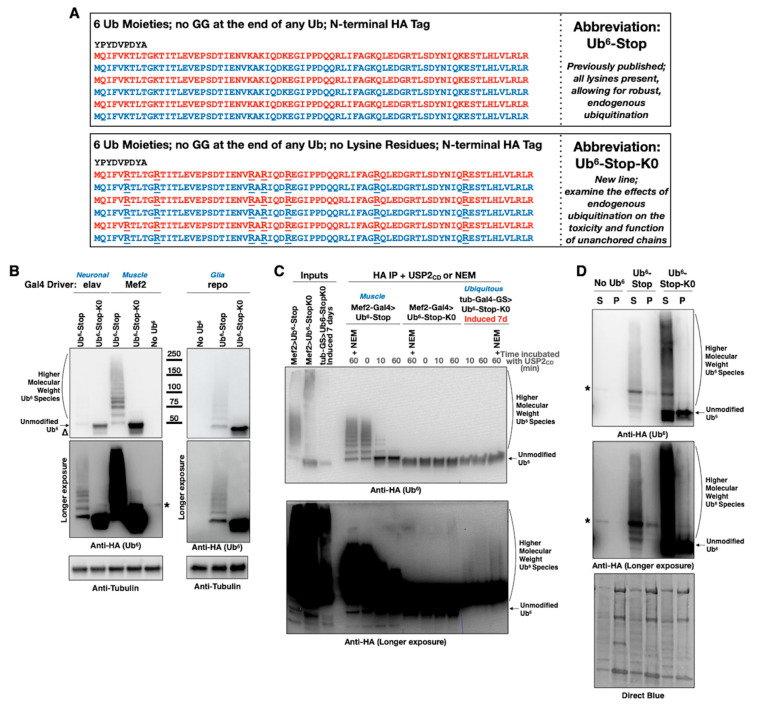
New transgenic *Drosophila* lines expressing non-cleavable, unanchored ubiquitin chains. (**A**) Amino acid sequence, abbreviation and purpose of HA-tagged ubiquitin chain transgenes. Neither chain can be conjugated onto another protein as it lacks a terminal “GG” motif that is required for isopeptide bond formation. Absence of internal “GG” motifs also ensures that they are not dismantled by DUBs [26]. All lysine (K) residues within Ub^6^-Stop-K0 were mutated into the similar, but non-ubiquitinatable, amino acid arginine (R) to prevent ubiquitination. (**B**) Western blots from flies expressing the noted transgenes in all neurons (elav-Gal4, adult lysates), muscle cells (Mef-Gal4, adult lysates), or all glial cells (repo-Gal4, pharate adult lysates). Delta symbol (Δ): signal underneath the main band of lysine-less Ub^6^ that we observe sometimes and could be a proteolytic fragment of the chain. (**C**) Western blots from stringently immunopurified, HA-tagged ubiquitin chains treated, or not, with the catalytic domain of USP2 for the indicated amounts of time. Mef2-Gal4 flies were one day old. Flies with tub-Gal4-GS driver were induced to express Ub^6^ for 7 days before being collected for protein extraction. (**D**) Western blots from soluble/pellet fractionation of flies expressing the noted ubiquitin chains in all muscle cells. Flies were one day old. The smear present in the Ub^6^-Stop-K0 samples comprises SDS-resistant species as a result of the buffer used in this protocol. As shown in Figure 1C, second lane from the left and USP2_CD_-treated lanes, similar smears from a different buffer and lysis protocol (Materials and Methods) are not collapsed by the addition of the DUB. Asterisks in panels: non-specific band detected by the anti-HA antibody.

**Figure 2 cells-09-01519-f002:**
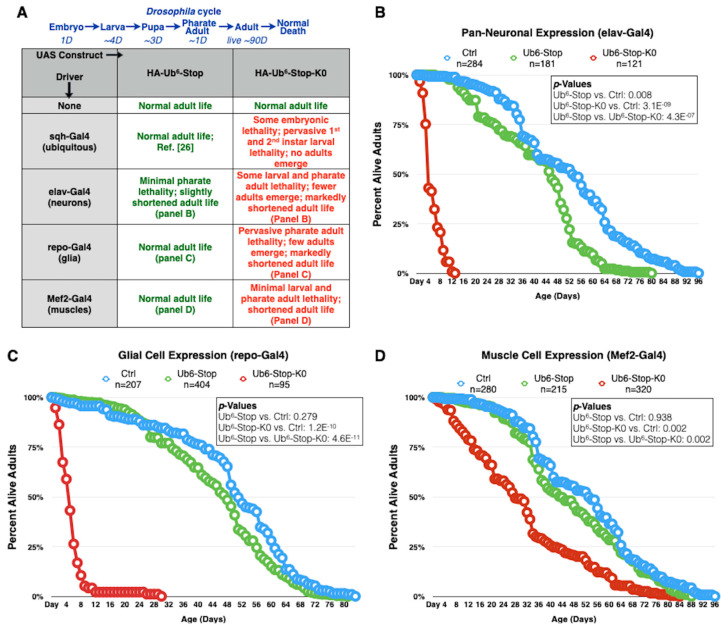
Lysine-less, non-cleavable ubiquitin chains are toxic in flies. (**A**) Summary of findings when the noted transgenes are expressed as indicated. “None” denotes flies that contained the indicated transgene without a Gal4 driver. Results are from crosses that were conducted simultaneously. A schematic of the fly life cycle, with the approximate number of days (“D”) spent in each stage, is provided. (**B**–**D**) Longevity assays of adult flies expressing Ub^6^ chains under the noted Gal4 drivers. “Ctrls” do not contain Ub^6^ transgenes, but have the Gal4 driver on the genetic background that we utilized to generate the ubiquitin chain-containing flies. ‘*p*’ values are from log-rank tests.

**Figure 3 cells-09-01519-f003:**
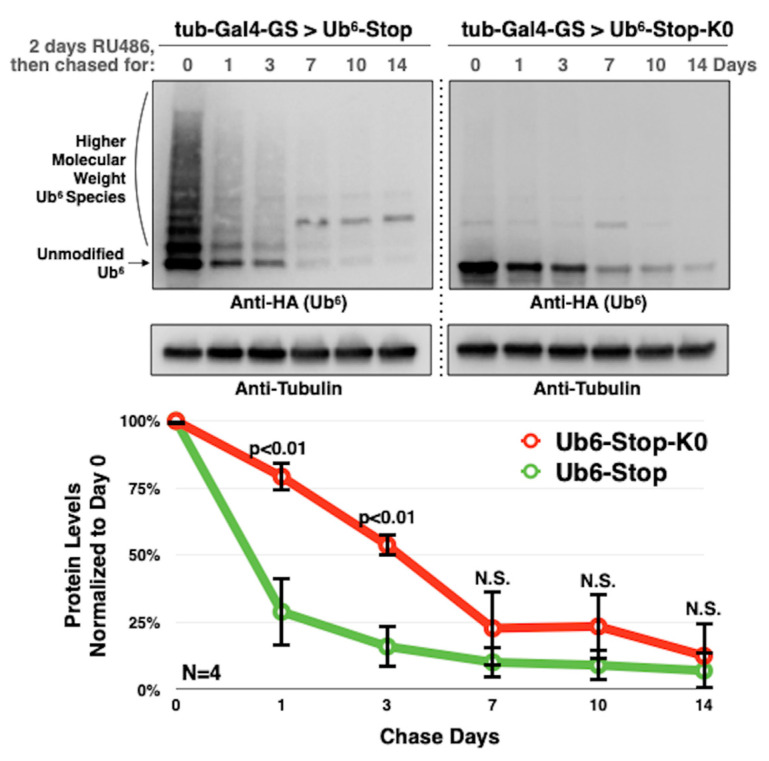
Decelerated turnover of lysine-less, non-cleavable ubiquitin chains in flies. Western blots from flies expressing the noted Ub^6^ transgenes under the RU486-dependent driver, tub-Gal4-GS for 2 days, and then “chased” in the absence of the inducer for up to 14 days. Whole-fly lysates. Graphs in the bottom are quantifications from images on the top and other independent repeats. The entire signal for each lane was quantified. Shown are means ± standard deviation. ‘*p*’ values are from Student’s *t*-tests comparing Ub^6^-Stop-K0 to Ub^6^-Stop of the same timepoint. “N.S.” denotes no statistical significance.

**Figure 4 cells-09-01519-f004:**
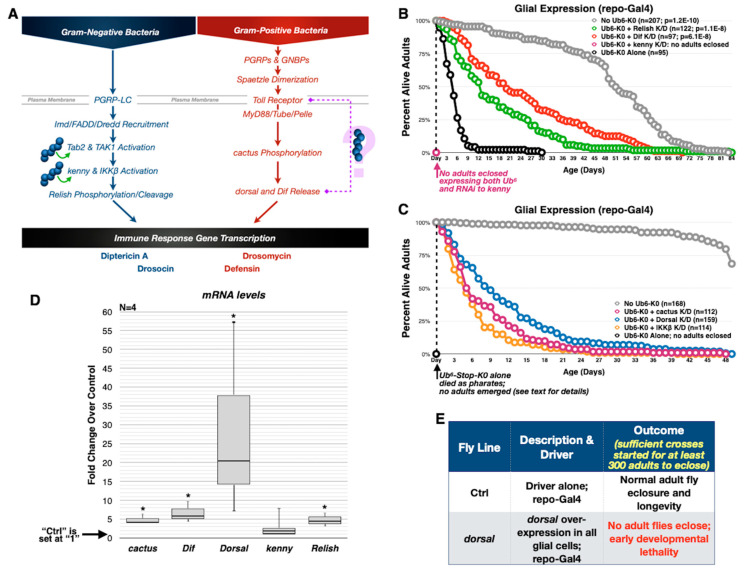
NF-kB pathway dysregulation when lysine-less, untethered hexa-ubiquitin is expressed in fly glia. (**A**) Summary of the IMD (left) and Toll (right) pathways in flies and potential roles for unanchored ubiquitin chains in them. (**B**,**C**) Longevity of adult flies expressing Ub^6^-Stop-K0 alone or in combination with RNAi-dependent knockdown of the noted NF-kB-related genes in fly glia. ‘*p*’ values are from log-rank tests. Longevity curves in panel B for “No Ub^6^-K0” and “Ub^6^-K0 Alone” are the same as in Figure 2C. Adult flies from experiments with RNAi lines were collected and aged simultaneously and supplement results from earlier set ups where no adults emerged from repo-Gal4 X Ub^6^-Stop-K0 crosses conducted for Figure 2C (all developing flies from those earlier crosses died as pharate adults). (**D**) Box-and-whisker plot of mRNA levels of the noted NF-kB-related genes by qRT-PCR from dissected heads of pharate adults expressing Ub^6^-Stop-K0 in glial cells, presented as fold change over controls. Statistics are from Student’s *t*-tests comparing levels of each gene from Ub^6^-Stop-K0 flies to “Ctrls” that contained the repo-Gal4 driver on the genetic background of Ub^6^-Stop-K0. Asterisks: *p* < 0.05. (**E**) Survival outcomes for flies that express exogenous *dorsal* in glial cells, in the absence of any Ub^6^.

**Figure 5 cells-09-01519-f005:**
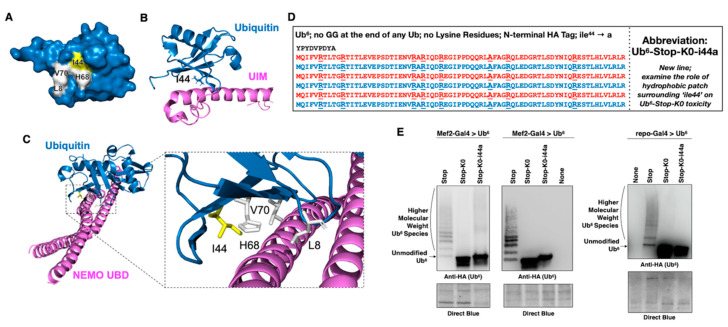
New transgenic, lysine-less ubiquitin chains with mutated isoleucine 44. (**A**) Surface representation of ubiquitin (PDB 1ubq) highlighting the structural location of the amino acids forming the hydrophobic patch involved in ubiquitin recognition (I44 in yellow, and L8, H68, and V70 in white). (**B**,**C**) Ribbon representation of ubiquitin (blue) displaying isoleucine 44 in the binding surface of different ubiquitin-interacting motifs in pink. (**B**, UIM; PDB 1yx5; **C**, the UBD domain of NEMO; PDB 2zvo). Inset highlights ubiquitin’s hydrophobic patch and the key position of isoleucine 44 (yellow). Images were generated using PyMol (Schrodinger, Inc., New York, NY, USA). (**D**) Summary of the new ile44a transgenic line that we generated, its abbreviation and purpose. (**E**) Western blots from flies expressing the noted transgenes in all muscle cells (Mef2-Gal4, adult lysates) or all glial cells (repo-Gal4, pharate adult lysates).

**Figure 6 cells-09-01519-f006:**
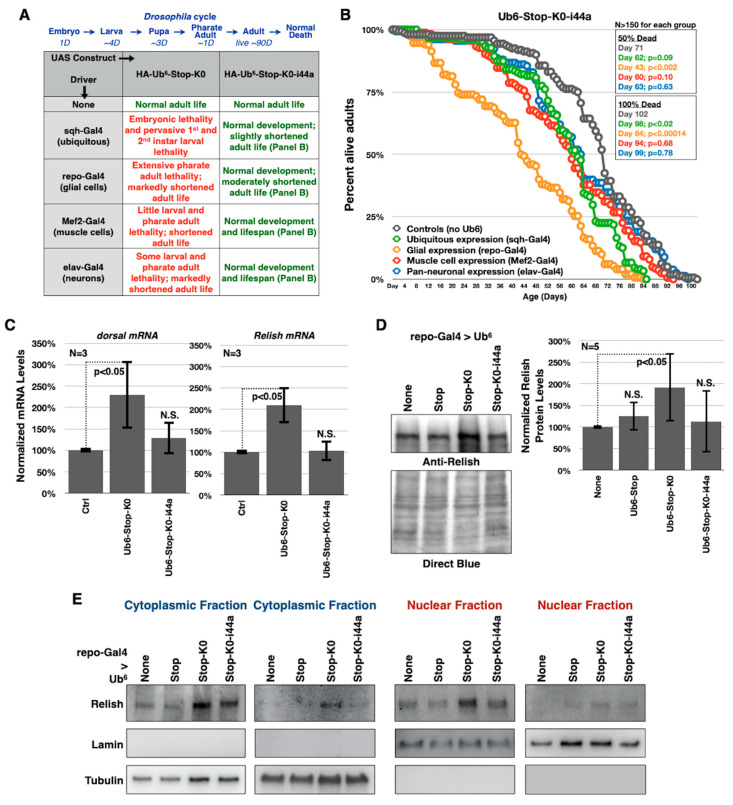
Isoleucine 44 mediates toxicity from free, linear, lysine-less ubiquitin chains. (**A**) Summary of findings when the noted transgenes are expressed as described. “None” denotes no Gal4 driver was present. Results are from crosses that were conducted simultaneously. (**B**) Longevity assays of adult flies expressing Ub^6^ chains under the noted Gal4 drivers. “Ctrls” do not contain Ub^6^ transgenes, but have the Gal4 driver on the genetic background that we utilized to generate the Ub^6^ chain-containing flies. ‘*p*’ values are from log-rank tests. (**C**) mRNA levels of the noted NF-kB-related genes by qRT-PCR in dissected heads of pharate adults expressing Ub^6^-Stop-K0 or Ub^6^-Stop-K0-i44a in glial cells, presented as fold change over controls without the Ub^6^ transgene. ‘*p*’ values are from Student’s *t*-tests comparing Ub^6^-expressing flies to controls. (**D**) Western blots from dissected pharate adult heads expressing the noted transgenes in all glial cells (left). Relish protein levels were normalized to direct blue and quantified (right). ‘*p*’ value is from Student’s *t*-tests comparing Ub^6^-expressing flies to controls that contain the driver on the same genetic background as ubiquitin chain-encoding flies. (**E**) Western blots from two independent cytoplasmic/nuclear fractionations of dissected pharate adult heads expressing, or not, Ub^6^ transgenes in glial cells. In (**C**,**D**), “N.S.” denotes no statistical significance comparing a specific Ub^6^ group to its respective control. Due to the temporary closure of our institution as a result of the COVID-19 pandemic, we were unable to obtain additional, independent samples for panel (**E**). We also encountered some difficulties with signal-to-noise ratio from the anti-Relish antibody for panel (**E**), but were unable to procure more reagents as a result of university closure.

**Figure 7 cells-09-01519-f007:**
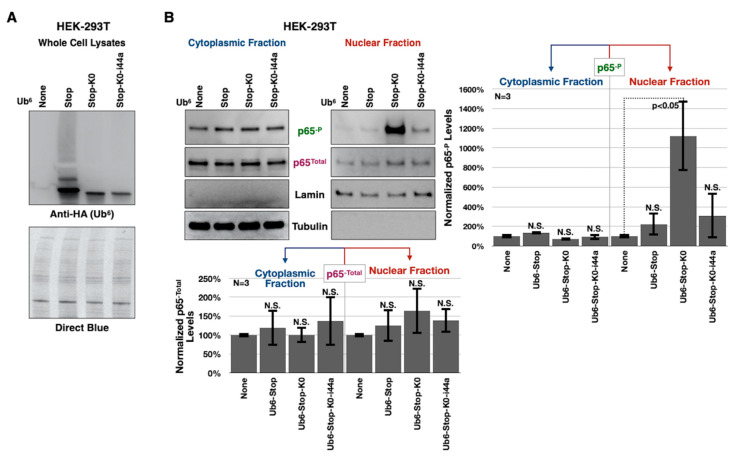
Isoleucine 44 mediates nuclear translocation of phosphorylated p65 in mammalian cells. (**A**) Western blots from whole-cell lysates of HEK-293T cells transfected with empty vector or Ub^6^ plasmids. (**B**) Western blots and quantifications from cytoplasmic/nuclear fractionation of HEK-293T cells transfected as in (**A**). For both (**A**,**B**), results are representative of three independent experiments. In panel (**B**), p65 protein levels were quantified, divided by the respective loading control (tubulin or lamin) for each fraction and normalized to controls lacking Ub^6^. Shown in histograms are means ± standard deviation. ‘*p*’ value is from Student’s *t*-tests comparing cells expressing Ub^6^ to non-Ub^6^-expressing controls. “N.S.” denotes no statistical significance comparing a specific Ub^6^ group to its respective control. p65^-P^: phosphorylated-p65. The COVID-19 situation also hampered our ability to conduct additional cell-based studies.

**Figure 8 cells-09-01519-f008:**
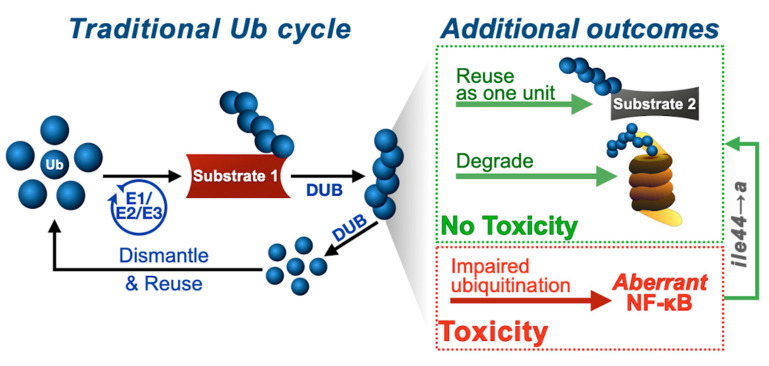
Summary of potential outcomes of ubiquitin recycling. “Traditional Ub cycle”: ubiquitin is attached to a substrate protein either as a monomer (not shown) or as a polymer. Poly-ubiquitin can be removed as a single unit by DUBs, dismantled into mono-ubiquitin, and then reused. “Additional outcomes”: based on our studies in *Drosophila* from earlier investigations and from this work, unanchored poly-ubiquitin can be reused as a single unit and become attached onto another protein [26], or can be degraded ([26] and this work). If handled as described thus far, unanchored chains are not detrimental (No Toxicity). However, if the ability of cells to reuse or to ubiquitinate free poly-ubiquitin as described here is hampered, unanchored chain toxicity results from cellular pathway perturbation, as with NF-kB processes evidenced here (Toxicity). Based on our data, this toxicity is reversed by an isoleucine 44 to alanine mutation that disrupts a critical interaction site on ubiquitin.

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
