# Peer review of "Isoleucine 44 Hydrophobic Patch Controls Toxicity of Unanchored, Linear Ubiquitin Chains through NF-κB Signaling"

_cells, 2020, doi:10.3390/cells9061519_

Round 1
Reviewer 1 Report
This work by Blount et al investigates how the presence of unanchored ubiquitin chains affects cell signaling and viability in Drosophila. Unanchored Ub chains are often considered per se toxic for cells, which a prior publication of the same authors (Sci Rep 2018) has contested, by showing physiological, non-toxic roles of unanchored chains in Drosophila. Why they are nevertheless sometimes toxic remained elusive. In this manuscript the authors deeper probed into the enigma of toxic vs non-toxic unanchored ubiquitin chains by generating several transgenic Drosophila lines. One expresses a linear ubiquitin chain (Ub6-Stop), which cannot be conjugated to targets and cannot be disassembled by DUBs (all C-terminal GGs lacking). This chain type was previously shown not to be toxic to flies (Sci Rep 2018). The other line expresses the same ubiquitin chain plus mutation of all internal K residues to R (Ub6-Stop-K0), which prevents it from being modified by endogenous ubiquitin chains, which was a hallmark of Ub6-Stop. Data presented here suggest that unanchored ubiquitin chains become toxic when they are linear and made inert to modifications by other Ub chains, potentially affecting clearance. Lingering linear chains appear to stimulate toxic NF-kB signaling, which can be reversed by mutation of Ile44.
This is a highly interesting study, which significantly expands our knowledge in an understudied field. The chosen model for investigation, although not physiological, is suitable to address the asked questions and certainly led to some compelling results. Most conclusions are supported by good quality data; however, I do have concerns regarding causation and Ub modification of Ub6-Stop-K0.
- To me the link between Ub Ile44 toxicity and NF-kB signaling is rather weak, and some supporting data could be added. For example, it is not clear what is the upstream target that leads to upregulation of NF-kB genes. Nemo and Tab2/3 are mentioned as potential candidates, but none are investigated. Also, they would not explain the effect on Toll signaling. The only link between Ile44 and NF-kB shown is the mRNA expression levels of NF-kB genes that return to ctrl conditions when Ile44 is mutated. This however could be just a consequence of healthier flies.
- In the same regard, no downstream NF-kB genes were assessed to support NF-kB overactivation. Upregulation of NF-kB proteins does not necessarily mean an effect on dependent gene expression. Is there nuclear translocation of NF-kB and what is the expression level of downstream NF-kB-dependent genes in K0 and K0 Ile44 mutant?
- Another concern is that data in Figure 1D contradict the notion that Ub6-Stop-K0 is not further modified by ubiquitin, since in the soluble fraction there is tons of high molecular weight smear observed. In all other figures looking at this, Ub6-Stop-K0 does not show the smear. Why? This needs to be clarified as the entire study is based on that.
In the following some other rather minor comments/concerns that should be addressed:
- Can the authors comment on the obvious difference in ubiquitin attachment pattern to Ub6-Stop in different cells (Figure 1B)? While in muscle there appears to be preferential attachment of 3-5 moieties, in neurons and glia there is mostly addition of one Ub molecule only. Would that influence the impact that these chains have on function of different cells and tissues? Could that explain data on survival of different lines (Ub6-Stop in neuron/glia vs muscle) in Figure 2?
- In the same regard: What explains the different effect on survival in different Ub6-Stop-K0 driver lines in Figure 2? Please elaborate. What would the graph for the global line look like?
- For the Drosophila life cycle in Figure 2: for non-Drosophila scientists it would help to include the days spent in each life stage in the description of timeline. Otherwise this is hard to follow in the graphs.
- Turn-over of Ub6-Stop-K0 was assessed in inducible global lines in adults (Figure 3), which may not reflect the situation in earlier life stages, when flies effectively die with the other drivers. This is a caveat that should be discussed.
- Relish and Dif single knockdown in conjunction with glial Ub6-Stop-K0 leads to about 50% rescue of flies, indicating that both Imd and Toll pathways contribute to toxicity. Would the effect be additive if both Relish and Dif were knocked down together? Would there be a 100% rescue?
- In Figure 4C, the degree of rescue cannot be determined, as WT flies are lacking. Please include. In the same Figure, please relabel genes affected to match with the text.
- What drives expression of these NF-kB genes (Figure 4D)? It is kind of surprising that all genes (except Kenny) are upregulated.
- The schematic in Figure 7 is rather confusing. I think it would make more sense if focused on free chains (minus steps 1-3), toxic vs non-toxic. Also, “current model” and “additional outcomes” are misleading, and should be removed.
- Finally, the title of the manuscript is rather ambiguous and could be improved.
Author Response
From the authors: We have addressed each point below within the current confines of the COVID-19 situation, which have led to the closing of both Wayne State University and Duke University. We hope that our revised work is now deemed satisfactory by Reviewer 1 for publication in Cells.
Comments and Suggestions for Authors
This work by Blount et al investigates how the presence of unanchored ubiquitin chains affects cell signaling and viability in Drosophila. Unanchored Ub chains are often considered per se toxic for cells, which a prior publication of the same authors (Sci Rep 2018) has contested, by showing physiological, non-toxic roles of unanchored chains in Drosophila. Why they are nevertheless sometimes toxic remained elusive. In this manuscript the authors deeper probed into the enigma of toxic vs non-toxic unanchored ubiquitin chains by generating several transgenic Drosophila lines. One expresses a linear ubiquitin chain (Ub6-Stop), which cannot be conjugated to targets and cannot be disassembled by DUBs (all C-terminal GGs lacking). This chain type was previously shown not to be toxic to flies (Sci Rep 2018). The other line expresses the same ubiquitin chain plus mutation of all internal K residues to R (Ub6-Stop-K0), which prevents it from being modified by endogenous ubiquitin chains, which was a hallmark of Ub6-Stop. Data presented here suggest that unanchored ubiquitin chains become toxic when they are linear and made inert to modifications by other Ub chains, potentially affecting clearance. Lingering linear chains appear to stimulate toxic NF-kB signaling, which can be reversed by mutation of Ile44.
This is a highly interesting study, which significantly expands our knowledge in an understudied field. The chosen model for investigation, although not physiological, is suitable to address the asked questions and certainly led to some compelling results. Most conclusions are supported by good quality data…
Response: We thank reviewer 1 for these encouraging details. We greatly appreciate the care that reviewer 1 put into reading and assessing our work, and the insightful points raised below. We believe that our work has emerged stronger and more accessible to a wider audience as a result of these comments. We hope that our replies, itemized below, address the issues raised satisfactorily.
We conducted limited, additional assays to address some points. We have new data in figure 6E, and a new figure 7, which help tie Ub6-Stop-K0 to increased nuclear levels of Relish (flies) and of phosphorylated-p65 (mammalian cells), as described further below. We were limited in our ability to conduct extensive work due to the COVID-19 shutdown of all non-key research activities - still, we managed to obtain additional data. We hope that these additional results and text revisions sufficiently and satisfactorily address all points.
Again, thank you for your care, input and guidance.
however, I do have concerns regarding causation and Ub modification of Ub6-Stop-K0.
- To me the link between Ub Ile44 toxicity and NF-kB signaling is rather weak, and some supporting data could be added. For example, it is not clear what is the upstream target that leads to upregulation of NF-kB genes. Nemo and Tab2/3 are mentioned as potential candidates, but none are investigated. Also, they would not explain the effect on Toll signaling. The only link between Ile44 and NF-kB shown is the mRNA expression levels of NF-kB genes that return to ctrl conditions when Ile44 is mutated. This however could be just a consequence of healthier flies.
Response: We would like to start our reply by highlighting that the primary thrust for this current work was to examine conditions under which free chains become toxic in vivo, which we think we have suitably demonstrated. We also believe that our work provides sufficient guidance towards future investigations to mechanistically dissect these observations.
As noted by both reviewers 1 and 3, the topic of free Ub chains is one of interest; we believe that much future work is required in this understudied aspect of Ub biology. Still, reviewer 1’s comments on mechanistic details are appreciated.
Under the current COVID-19 situation, it will take many months to approach any type of mechanistic detail that reviewer 1 is referring to. With this in mind, we hope that the following modifications to the text sufficiently address the point above in the Discussion (bold type is added text):
“It remains to be resolved which proteins recognize and interact with Ub6-Stop-K0. For example, upstream elements that lead to upregulated NF-κB component levels in the presence of lysine-less, linear poly-Ub need to be identified. Still, we infer that these elements likely depend on interacting with the ile44 hydrophobic patch, since mutating this amino acid residue markedly reverses Ub6-Stop-K0 toxicity in intact flies. Our results highlight NF-κB components as key players in toxicity from ubiquitination-resistant, linear chains.…”
Insofar as the Toll pathway is concerned, we have made the following addition in the Discussion, shown here in bold type (underlined text pertains to a subsequent point, further below):
“Unlike for the Imd pathway, where evidence from mammalian orthologues and functional assays of fly proteins indicate a role for free Ub chains, it is unclear whether unanchored poly-Ub is involved in the Toll pathway. Toll signaling begins when gram-positive bacteria or fungi are detected by extracellular peptidoglycan recognition proteins and glucan-binding proteins, triggering a cascade that results in the processing of the Toll ligand Spaetzle to facilitate its binding to the Toll receptor [72-74]. After activation, the Toll receptor binds the adaptor MyD88 and recruits Tube and Pelle to phosphorylate cactus [75], ultimately activating dorsal and Dif [76] and driving the transcription of antimicrobial peptides like Drosomycin and Defensin (figure 4A and [77]). Insofar as we know, our data that free, linear poly-Ub increases levels of cactus and dorsal are the first to link linear chains to the Toll pathway, thus merging gram-negative and gram-positive/fungi, NF-κB-centered processes under the regulation of linear Ub chains. Which proteins serve as "receptors" that recognize lysine-less, linear poly-Ub and lead to activation of the Toll pathway? Studies with unbiased approaches and targeted genetics will be necessary to uncover these details and to also identify the downstream genes that are impacted by free, lysine-less Ub chains for both the Imd and Toll pathways. Based on increased nuclear levels of relish and p65 proteins in flies and mammalian cells expressing Ub6-Stop-K0, there is good reason to predict a genetic response to these transcription factors.
We believe that our interpretation of our overall results is consistent with our model that ile44 is important for NfKB-dependent toxicity from linear chains in flies. For the overall scope of the current manuscript, we hope that reviewer 1 agrees with our determination that we have accomplished the major task of understanding when free Ub chains can become toxic in vivo and have provided guiding mechanistic insight for future work.
- In the same regard, no downstream NF-kB genes were assessed to support NF-kB overactivation. Upregulation of NF-kB proteins does not necessarily mean an effect on dependent gene expression. Is there nuclear translocation of NF-kB and what is the expression level of downstream NF-kB-dependent genes in K0 and K0 Ile44 mutant?
Response: As mentioned in the manuscript, current fly tools for the type of work mentioned here are unreliable. We have tried many antibodies for various Nf-KB components, and, if helpful to the Reviewer, can provide them to see how unreliable they are, except for Relish. To address this concern in the revision, we added the last sentenced, underlined, in the prior point.
To address the question of nuclear translocation of NF-kB, we added additional data. Figure 6 now includes cytoplasmic and nuclear fractionation from flies to show that more Relish protein is present in the nuclei when Ub6-Stop-K0 is expressed, but not when ile44 is mutated (new panel 6E). We also added cytoplasmic/nuclear fractionation assays using mammalian, HEK-293T cells that show that the levels of an activated NF-κB protein (phosphorylated-p65) are dramatically increased in the nuclei of cells transfected with Ub6-Stop-K0 (new figure 7). We hope that the additions to the text below, in bold, adequately address reviewer 1’s concern:
“Because NF-κB transcription factors translocate to the nucleus when activated, we performed subcellular fractionation to visualize nuclear Relish protein levels in heads dissected from pharate adults expressing Ubin glial cells. We observed that more Relish is present in the nuclei of Ub-Stop-K0 pharate adults, compared to controls that are genetically similar, but lack Ub(figure 6E). Nuclear Relish levels in the presence of either Ub-Stop or Ub-Stop-K0-i44a expression also appear lower than those in the presence of Ub-Stop-K0, indicating the importance of the chains’ susceptibility to ubiquitination and ile44. Collectively, these results lead us to conclude that ubiquitination-resistant, free, linear Ub chains can be highly toxic in flies at least in part due to NF-κB signaling that is dependent on the ile44-centered hydrophobic patch on Ub.
Due to a lack of quality antibodies for NF-κB components in flies, we performed subcellular fractionation experiments using cultured, HEK-293T human cells to validate our findings in figure 6E. In humans, p65 is an NF-κB transcription factor whose phosphorylation at ser536 controls its nuclear translocation and activity [59]. To examine p65 translocation, we transiently transfected HEK-293T cells with the Ub6 constructs and 24 hours later performed subcellular fractionation and western blotting. Expression of the various Ub6 constructs in HEK-293T cells leads to ubiquitinated species for Ub6-Stop, but not so for lysine-less variants (figure 7A). As shown in figure 7B, Ub6-Stop-K0 expression leads to markedly more endogenous phosphorylated-p65 in the nucleus compared to cells transfected with empty vector; the ile44 mutation again abrogates this effect. There was no statistically significant difference in the total p65 levels in the cytoplasm or nucleus (figure 7B). Based on the increased nuclear translocation of activated p65, it is likely that Ub6-Stop-K0 induces the transcription of downstream NF-κB response genes; additional studies are needed to determine the precise outcomes. These results reinforce a role for ubiquitination-resistant, unanchored, linear poly-Ub in NF-κB signaling, notably in a human cell line.”
- Another concern is that data in Figure 1D contradict the notion that Ub6-Stop-K0 is not further modified by ubiquitin, since in the soluble fraction there is tons of high molecular weight smear observed. In all other figures looking at this, Ub6-Stop-K0 does not show the smear. Why? This needs to be clarified as the entire study is based on that.
Response: This is a good point that we should have clarified in the prior version of this work. We are confident that the smears are SDS-resistant Stop-K0 that results from the buffer used and the protocol itself. We used a mild buffer here, NETN, which is necessitated by the protocol. We now clarify this in the legend of figure 1, as pasted below. Also, reviewer 1 will note that while there is a smear above Ub6-Stop-K0 in this panel, it does not have the type of banding that can be seen with the Ub6-Stop transgene, which results from the ubiquitination of the latter construct.
In the legend of figure 1, bold text has been added: “(D) Western blots from soluble/pellet fractionation of flies expressing the noted ubiquitin chains in all muscle cells. Flies were one day old. Asterisks in panels: non-specific band detected by the anti-HA antibody. The smear present in the Ub6-Stop-K0 samples comprises SDS-resistant species as a result of the buffer used in this protocol. As shown in figure 1C second lane from the left and USP2-treated lanes, similar smears from a different buffer and lysis protocol (Materials and Methods) are not collapsed by the addition of the DUB.”
In the following some other rather minor comments/concerns that should be addressed:
- Can the authors comment on the obvious difference in ubiquitin attachment pattern to Ub6-Stop in different cells (Figure 1B)? While in muscle there appears to be preferential attachment of 3-5 moieties, in neurons and glia there is mostly addition of one Ub molecule only. Would that influence the impact that these chains have on function of different cells and tissues? Could that explain data on survival of different lines (Ub6-Stop in neuron/glia vs muscle) in Figure 2?
Response: This is an astute observation from Reviewer 1. We have made the following modification in the Results section pertaining to this observation:
“At this point, it is interesting to note that the precise type of ubiquitination laddering above the main band of Ub6-Stop differs among different tissues. In muscle cells, there appears to be a higher preponderance of >Ub1 addition (Ub3-5, middle portion of the blots in figure 1B; also see figure 5E) compared to neuronal and glial cells, whose signal is stronger closer to the main band. The reasons for these differences and their potential physiological consequences are presently unknown.”
- In the same regard: What explains the different effect on survival in different Ub6-Stop-K0 driver lines in Figure 2? Please elaborate. What would the graph for the global line look like?
Response: We have modified the text as highlighted below in bold type in the Results section to address this point:
“In essence, with every Gal4 driver used, Ub6-Stop-K0 leads to markedly reduced lifespan compared to both Ub6-Stop flies and control flies that are genetically similar but do not express either Ub6 (figure 2). Clearly, different drivers lead to variable toxicity from ubiquitination-resistant, linear poly-Ub. Among the tissues examined, muscle cells are least impacted. This is unlikely to be directly due to differences in relative amounts expressed by the different drivers, since we showed before that muscle cells express Ub6 transgenes highly robustly [26]. More likely, different tissues handle and tolerate these species in different ways, leading to the observed variation in toxicity. These findings highlight the need to investigate Ub-dependent pathways in a tissue-specific manner to understand the full scope of their roles and importance in vivo. Collectively, these data point to marked toxicity from ubiquitination-resistant, untethered Ub6 species in Drosophila, unlike the ubiquitination-prone counterpart.”
The global driver (sqh-Gal4) does not yield any adults to include in longevity curves, as shown in figure 2A.
- For the Drosophila life cycle in Figure 2: for non-Drosophila scientists it would help to include the days spent in each life stage in the description of timeline. Otherwise this is hard to follow in the graphs.
Response: Very good point. We have done as recommended in both figures 2 and 6.
- Turn-over of Ub6-Stop-K0 was assessed in inducible global lines in adults (Figure 3), which may not reflect the situation in earlier life stages, when flies effectively die with the other drivers. This is a caveat that should be discussed.
Response: We have done as reviewer 1 recommends. The text has been amended as pasted below, with the modifications highlighted in bold type:
“Flies heterozygous for tub-Gal4-GS and either form of Ub6 were reared on regular fly food until they emerged from their pupal cases, at which point they were switched to RU486 food for 2 days to induce Ub6 transgene expression (the pulse). Adults were then switched to regular media to halt additional Ub6 production and were flash-frozen periodically to assess Ub6 disappearance by Western blotting (the chase). We selected an inducible, ubiquitous Gal4 driver and focused on adult flies to gain a general perspective on the turnover of these Ub species, aware of the fact that degradation rates may differ in a tissue- and developmental stage-dependent manner. Nonetheless, this approach provides us with critical insight into the handling of ubiquitinatable and ubiquitination-resistant, linear poly-Ub at the level of the whole organism. Figure 3 summarizes our findings.”
- Relish and Dif single knockdown in conjunction with glial Ub6-Stop-K0 leads to about 50% rescue of flies, indicating that both Imd and Toll pathways contribute to toxicity. Would the effect be additive if both Relish and Dif were knocked down together? Would there be a 100% rescue?
Response: We wish that we could conduct this experiments, but the genetics of it are impossible with current tools on hand. We would need to generate new RNAi lines for either or both genes to accomplish this. This will be impossible for many months. However, we believe that our current data still provide insightful evidence into genetic mechanisms involved.
- In Figure 4C, the degree of rescue cannot be determined, as WT flies are lacking. Please include. In the same Figure, please relabel genes affected to match with the text.
Response: We have done as recommended by Reviewer 1 for labels and control flies in the text and in figure 4C.
- What drives expression of these NF-kB genes (Figure 4D)? It is kind of surprising that all genes (except Kenny) are upregulated.
Response: We hope that our response to point 1 above provides a sufficient reply to this point.
- The schematic in Figure 7 is rather confusing. I think it would make more sense if focused on free chains (minus steps 1-3), toxic vs non-toxic. Also, “current model” and “additional outcomes” are misleading, and should be removed.
Response: We made a few modifications to the figure to help the eye transition better and used different labels and different colors. We retained steps 1, 2 and 3 to help the reader who might not be a Ub aficionado. We replaced the “current” and “additional” labels and adjusted the text to reflect these changes. We will be happy to make further changes if reviews 1 still finds this figure confusing.
- Finally, the title of the manuscript is rather ambiguous and could be improved.
Response: We changed the title to: “Isoleucine 44 hydrophobic patch controls specific cases of toxicity from linear, unanchored ubiquitin chains in vivo”.
Reviewer 2 Report
The text is very well written and has clarity.
It is important that the same care is taken in the presentation of figures too!
Author Response
Response: Thank you for your appreciation of our work.
Reviewer 3 Report
In this study, Blunt and colleagues investigate the potential toxicity deriving from the expression of free linear poly-ubiquitin chains in Drosophila melanogaster. The authors previously showed that untethered poly-Ub chains are not intrinsically harmful in flies, however other studies reported toxicity from untethered ubiquitin chains. To examine under which circumstance the expression of poly-ubiquitin chains is toxic, the authors transgenically expressed in flies two Ub6 constructs, i.e. Ub6-Stop and Ub6-Stop-K0, which are both DUBs-resistant although the latter is devoid of lysine residues and thus it is ubiquitination-resistant. The authors show that Ub6-Stop-K0 is more stable and its expression in flies results in a markedly reduced lifespan when compared to Ub6-Stop. The toxicity of lysine-less, unanchored poly-Ub relies on aberrant NF-κB signalling and is dependent on the Ile44 residue located at the center of the hydrophobic patch which is involved in ubiquitin recognition by ubiquitin-binding domains.
I am encouraging the publication of this work because it provides new interesting insights into ubiquitin functions and recycling. The experiments are presented in a straightforward and logical fashion. Most data support the authors’ conclusions. I believe that this study represents a solid starting point for further work on the regulation of free untethered poly-Ub chains in more complex organisms and under various physiological/pathological conditions.
Author Response
Response: We thank reviewer 3 for the thorough read, these encouraging comments and for supporting our work.
Round 2
Reviewer 1 Report
This reviewer appreciates the thoroughness with which the authors have responded to the major and minor comments. With the new data and text, the NF-kB involvement in free-Ub chain toxicity is sufficiently supported within the scope of the manuscript. All other points have been addressed as well.
My only remaining concerns regard the schematic in Figure 8 and the title of the manuscript, which I feel are both still not 100% clear.
For Figure 8 please consider these suggestions:
- The numbers 1-5 are confusing, as they indicate a sequential action, which is not the case for 4+5. I suggest to remove them, or think of another way to indicate the steps.
- Steps 4+5 are mutually permissive and possible. This is not clearly outlined in the schematic. Both arrows to toxicity point away from 4+5, which insinuates a toxicity of these steps where there is none.
- In general, the rightmost panel “toxicity” is not very informative. The “Yes” and “No” should be omitted, as this is confusing with wording where “Yes” leads to NO toxicity. And the I44A position towards more toxicity is not ideal.
- The separation of the schematic into the three sections (traditional Ub cycle, additional possibilities and toxicity) is misleading. In my opinion additional possibilities should be renamed, the current name is rather vague. The toxicity header should be removed and contents combined under the renamed section 2.
As for the title, consider a change to: Isoleucine 44 hydrophobic patch controls toxicity of linear, unanchored ubiquitin chains in vivo.
